# Moderately Prolonged QTc in Computer-Assessed ECG, Random Variation or Significant Risk Factor? A Literature Review

Jan Hysing [1,*], Charlotte Gibbs [1], Øystein Lunde Holla [1], Jacob Thalamus [1] and Kristina H. Haugaa [2]

1 Telemark Hospital Trust, N-3710 Skien, Norway
2 Oslo University Hospital, N-0424 Oslo, Norway
* Correspondence: hysj@sthf.no

**Abstract:** Most ECGs in European hospitals are recorded with equipment giving computer measured intervals and interpretation of the recording. In addition to measurements of interval and QRS axis, this interpretation frequently provides the Bazett's-corrected QTc time. The introduction of computer-corrected QTc revealed QTc prolongation to be a frequent condition among medical patients. Nevertheless, the finding is frequently overlooked by the treating physician. The authors combine experience from a local hospital with a review of the current literature in this field in order to elucidate the importance of this risk factor both as congenital long QT syndrome and as acquired QT prolongation.

**Keywords:** QT prolongation; long QT syndrome; genetic testing; inherited arrhythmia

## 1. Introduction

The clinical features of long QT syndrome were first published 1957 by Jervell and Lange Nielsen [1]. They described a family with six children of which four were deaf, and three children died of sudden cardiac arrest at the age of four, five and nine years, after a period with frequent fainting attacks. They all had a prolonged QT interval on their ECGs, and the publication documented that the QT intervals were prolonged even after correction for heart rate [1,2]. Some years later, two pediatricians, Romano and Ward, both on an individual basis published case reports of children with long QT intervals and syncope without deafness. Later research has revealed that the cases presented by Jervell and Lange Nielsen most likely had a homozygote defect of the KCNQ1 gene, whereas the cases presented by Romano and Ward were heterozygote for these defects and therefore developed less severe disease. In the following decades, science in this field progressed mainly through case history description and family register research [3–5]. In 1975, the difference between congenital (cLQTS) and acquired long QT syndrome (aLQTS) was acknowledged [3,6]. The clinical picture of the inherited syndrome was well established as Schwartz published the first diagnostic score in 1985 [4]. The autosomal dominant heredity of the syndrome was established [4], although a substantial variation in penetrance was acknowledged. Furthermore, the therapeutic approach with beta blockade treatment and the non-pharmacological approach with left stellate ganglion ablation was established.

## 2. Molecular Genetic Studies

In 1995-96 the three genes on chromosomes 11, 7 and 3 associated with the cLQTS were identified and the DNA sequence defined [7–9]. The membrane protein encoded by KCNH2 is illustrated in Figure 1 from reference [7]. The genes KCNQ1 and KCNH2 were reported to code for the slow (IKs) and for the rapid (IKr) potassium channels. It was described that the mutations in LQTS patients resulted in errors in IKs, IKr and in the sodium channel, SCN5A, and the defects were thereby explained on the cellular biology level. The errors in the three ion channels could be linked to the clinical syndromes LQT1, LQT2 and LQT3.

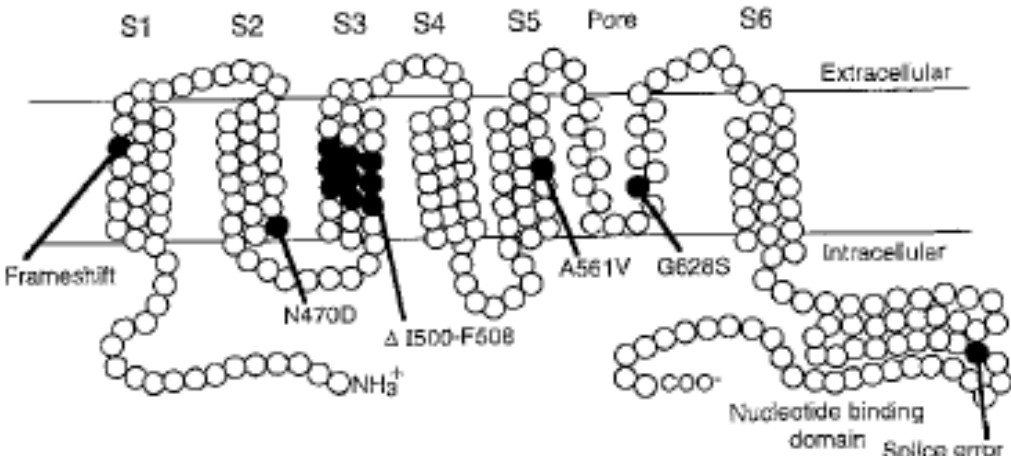

**Figure 1.** From reference [7], one of several illustrations from the series of scientific publications revealing the molecular knowledge on LQT mutations and ion channelopathies. Reproduced with permission from Elsevier Publisher.

The genes leading to errors in these channels contribute to 75% of all clinical cases, and are now characterized as major LQTS genes [10]. Efforts to explain the remaining heritability of LQTS led to the reporting of numerous new genes linked to clinical syndromes labeled LQT4 to LQT17. These genes have now in recent years been characterized as minor LQTS genes. The gene–disease associations for many of these minor genes seems to be questionable, and their effects seem easier to explain in a polygenic model. In polygenic models, common gene variants may also contribute to development of rare clinical disorders. These polygenic models also give support to the term "reduced repolarization reserve" originally introduced by Roden [11] in 1998. The model is not strictly scientifically defined, but it elucidates that concealed changes in cell membrane capacity, medication, electrolyte disturbances, cytotoxic effects, organ failure and central nervous stimulation all together add up, and may result in prolonged QT time and eventually torsade de point ventricular arrhythmia. This theory may help explain that pharmaceutical products, which in most patients are harmless, in others, given heritage or electrolyte disturbances, might result in life-threatening ventricular arrhythmias. An example given on rare adverse effects through QT prolongation was shown in studies on azithromycin, which usually is a well-tolerated antibiotic widely used. These studies indicated an increased risk of sudden cardiac death in 1/21 000 prescriptions compared with amoxicillin [12], highlighting the importance of QT awareness.

## 3. Mapping of Gene Penetrance Variation

Variable penetrance of inherited LQTS was described as early as 1975 and was later confirmed [3,13]. Not all patients with QT prolongation experienced ventricular arrhythmias. Furthermore, some siblings without QT prolongation on ECG experienced similar symptoms with fainting attacks and sudden death as family members with QTc prolongation.

The distribution of QTc measurements in mutation carriers and healthy controls as illustrated in Figure 2, was published 2007 presenting some of the complexity in this field [14].

The results from an important registry published in 2018 by Mazzanti and coworkers, elucidated this topic further as illustrated in Figure 3, from reference [15]. The study collected data from a large family registry of 1710 individuals from 812 families with LQTS.

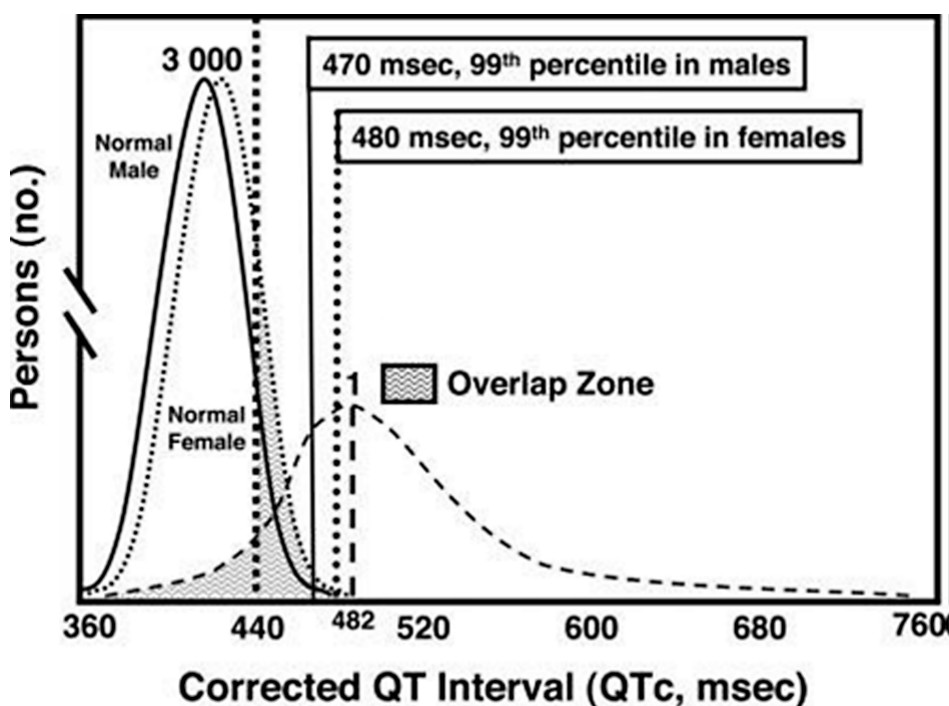

**Figure 2.** From reference [14], illustrates the distribution of QTc intervals in a cohort of patients with known LQT mutations, compared with male and female probands without known QT mutations. Reproduced with permission from Wolter Kluwer Health Inc.

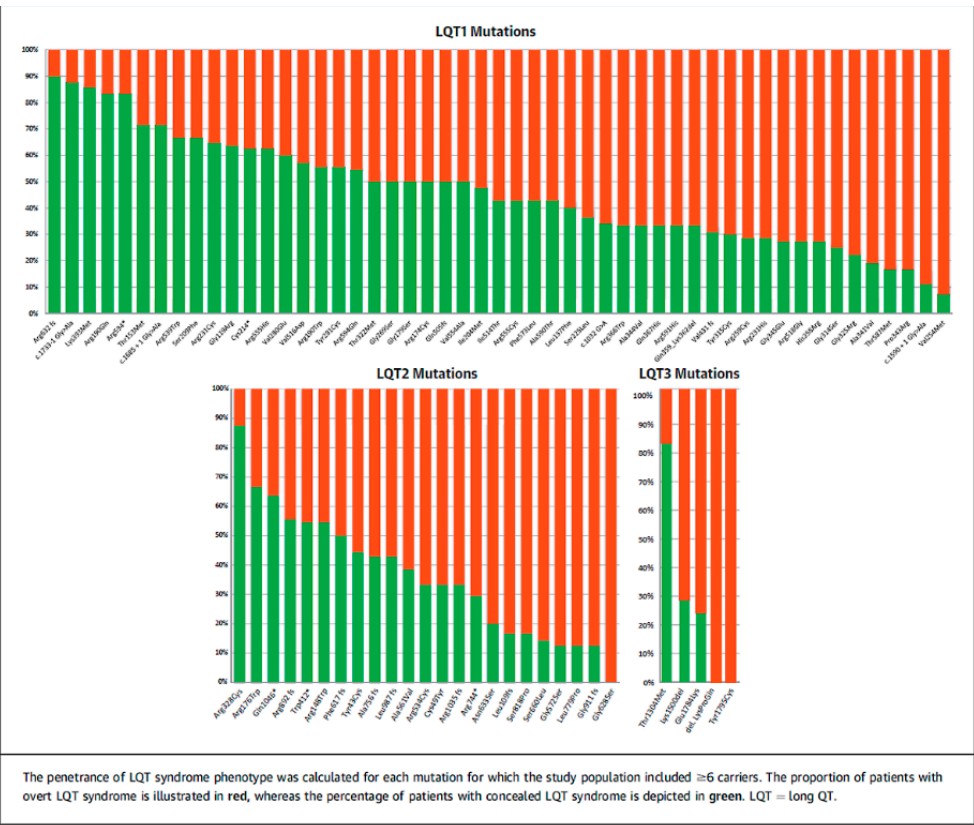

**Figure 3.** From reference [15] illustrates the distribution between concealed (green fraction) and overt (red fraction) in 53 mutations with LQT1, 23 mutations with LQT2 and 5 mutations in LQT3. Reproduced with permission from Elsevier Publisher.

Genetic analyses were performed and all together 79 genotypes were identified from the variants LQT1, LQT2 and LQT3. It was documented that 42% of all mutation carriers had a concealed LQTS with QTc < 460 ms. Substantial variation was found in penetrance between the various mutations. Some mutations had a penetrance of less than 15%, leaving more than 85% of the patients with a concealed LQTS. Other mutations had a penetrance up to 100%. The registry study followed patients for life-threatening arrhythmic events (LAE) and reported a 0.09% annual rate of LAE in patients with concealed LQTS, which was 12% of the risk in overt LQTS.

Furthermore, the registry study could document the relationship between increasing QTc measurements and risk of LAE. For each of the three syndromes, LQT1, LQT2 and LQT3 a correlation was found with increasing risk of LAE related to increasing QTc values. The 5-year risk for LAE increases by LQT1 from 1.3%, by QTc duration 461–470, up to 5.1% by QTc duration > 560 ms. Analogous to this, LQT3 patients with QTc duration 461–470 had a 5-year risk of LAE at 3% increasing to 12% with QTc duration of >560. Penetrance seems correlate to QTc and risk of LAE. A part of this variation may be explained on the basis of molecular biology. For the gene, KCNQ1, changes in the amino acid sequence in segment S6 significantly increase the risk of arrhythmic events related to other variants [16]. This broad spectrum in risk for LAE in LQTS, forms the backdrop for the "reduced repolarization reserve" seen in clinical practice.

## 4. Diagnostics and Treatment

If untreated, index patients with inherited long QT syndrome have a 10-year mortality risk of approximately 50% [4]. It is therefore very important to make a correct diagnosis and start treatment. A correct diagnosis must be based on QTc prolongation in ECG, clinical history and family history (Schwartz score) as well as genetic testing for major LQTS genes. In additional to avoidance and precautions, treatment with Nadolol reduces the risk of LAE and sudden death by 60%.

Several papers have tried to estimate the incidence of congenital as well as acquired LQTS in various cohorts [17–19]. In Italy, the incidence of congenital LQTS was estimated to 1/2500 healthy births [20]. An interesting approach published by Berge et al. [21], counted the number of homozygote Jervell–Lange-Nielsen patients in Norway. From the limited number of homozygote patients, the incidence of the heterozygote mutation carriers in Norway was estimated to be one in 317 births according to the Hardy–Weinberg law of random mating, by this procedure an incidence of 1/317 was found. Congenital LQTS tend to be exposed clinically in childhood or adolescence [22]. Pediatric ECGs are a special challenge because of the physiological tachycardia in small children. ECG with a heart rate of more than 100 should be judged with caution related to the substantial Bazett's correction. Acquired QTc prolongation is more often found in patients after 40 years of age and predominantly after the age of 60 years.

## 5. LQTS Incidence Is Highly Dependent on the Cohort Studied

As for acquired LQTS, the frequency is highly dependent on which cohort is being studied, and on how we define the limit for QT prolongation. In 2007, ECG reference ranges for intervals were derived from a cohort of 79,743 healthy volunteers. The 95% confidence intervals for QTc were found to be between 361 and 457 ms [23]. However, this was a pure statistical definition of normal range. It was also found that women on average had a 10 ms longer QT interval than men [23]. In concordance with this, the European guidelines defined QTc > 480 as abnormal and associated with increased risk of arrhythmia [24]. Having defined the limits, it remains a challenge for clinicians to recognize the diagnosis of QTc prolongation. Unless the heart rate is 60/min, the QT interval has to be adjusted to the heart rate. Probably for historical reasons the most used method is the Bazett's formula, and this is the method recommended in the HRS/EHRA/APHRS expert consensus statement on the diagnosis and management of patients with inherited primary arrhythmia syndromes [25]; however, Fredricia's and several other formulas are also used.

A challenge in the validation of these formulas is the lack of "gold standard", several studies indicate that Bazett's formula might overcorrect QTc with pulse rates above 90–100, and that the Fridericia and Framingham correction formulas perform better if the heartrate is >90/min [26,27]. A significant improvement in QT interval heart rate correction was achieved by the computer-corrected QT measurement, including user optional choice of correction formula, provided by several equipment vendors. Before the introduction of the computer-corrected ECG, inaccurate electrographic interpretation was documented on most continents [28]. The computer algorithms performing QTc calculation have been through proper validation and have been proven to perform well [29]. Certainly, the computer-corrected ECG must be manually checked for electromechanical noise and arrhythmias such as extrasystoles and bigeminy. Before making the diagnosis of congenital LQTS the ECG should be checked by a cardiologist. The frequent occurrence of QT prolongation found in medical departments, and especially in intensive care units may emphasize this. It has been estimated that LQTc prolongation may be seen in one out of five patients in intensive care unit [18]. The use of QT prolonging agents, and electrolyte disturbances, are key elements in this group of patients. So, the incidence of acquired QT prolongation as well as congenital LQTS are in some cohorts so high that genetic testing seems warranted [30].

## 6. Materials and Results from a Community Hospital

After introducing computer-corrected ECG readings in our hospital 18 years ago, the previously rare diagnosis of prolonged QT occurred several times a week. The question was raised on how we should respond to this. Was this a random finding just like a potassium or creatinine value just above the normal limit? In order to find out more about frequency, genetics and prognoses of prolonged QTc time we conducted a survey [31,32]. According to recommendations in the HRS/EHRA/APHRS expert consensus statement, we used the Bazett's correction formula. Of the 1855 patients with one or several ECGs diagnosed with QTc > 500ms, 324 (17%) had to be excluded, due to incorrect diagnosis. The main reason for errors were: electromechanical noise n = 152, inclusion of the P wave or U wave in the calculation of the QT interval n = 47, ventricular bigeminy n = 46, atrial fibrillation n = 19, short supraventricular arrhythmia n = 13 or other reasons n = 47. In a random sample of 200 ECGs with automatically measured QTc < 500ms, we found no ECGs with manually measured QTc $\geq$ 500 ms. The manually measured median QTc was 430ms (range 339–499) vs. automatically measured median QTc 434 ms (range 346–496). The correlation between the manually and the automatically measured QTc values was 0.91 ($p < 0.001$). Shortly summarized we found the phenomenon to be frequent in a community hospital. Any increase in QTc above normal value proved to be a risk marker, and patients with QTc > 500 ms had a less favorable 3-year survival [31], as illustrated by the Kaplan-Meyer survival plots in Figure 4 from reference [31].

The majority of the cases were acquired LQTS. We performed genetic testing in 475 patients with prolonged QTc on ECG and we found a genetic mutation in 6% of the patients tested. Most of the patients had electrolyte disturbances or they had QT prolonging medication. In our hospital, the occurrence of prolonged QT time was rarely mentioned in the medical records (only in 12% of the cases), and it was unclear to what extent the physicians tried to reduce the QT prolongation.

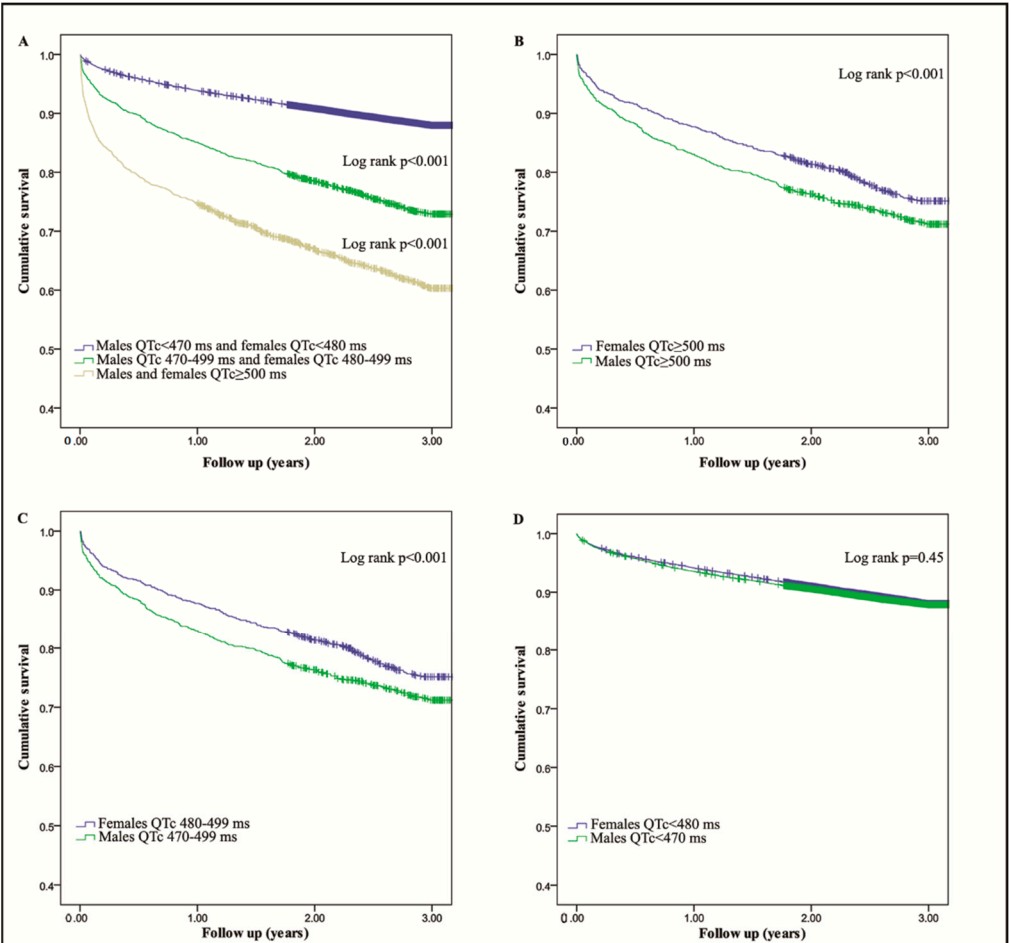

**Figure 4.** From reference [31]: the panels (**A–D**), illustrate 3-year survival curves for 1531 male and female patients with QTc > 500 ms versus controls, 2099 male and females with 500 > QTc > 470–480, and for 43,893 male and female without QTc prolongation. Reproduced with permission from Oxford University Press.

## 7. Discussion

Several large hospital clinics have implemented computerized clinical decision support systems for "QT alert", and it seems from published papers that these systems lead to reduction in prescription of QT prolonging drugs in risk patients [33,34]. One of the challenges for deciding physicians is the fact that many clinical conditions with minor risks frequently lead to QT prolongation, e.g., left ventricular branch block (LBBB), left ventricular hypertrophy from hypertension, surgery and non-ST elevation myocardial infarction, thus confusing the physician about appropriate actions. For these examples, some important information seems documented: the left bundle branch block, especially combined with congestive heart failure, but even without heart failure is a significant but rather weak risk factor for early mortality [35,36]. There are no validated methods to estimate the risk for torsade de pointes in QT prolongation in the presence of bundle branch block. Recommendations propose to subtract 50 ms from the QT interval before correction. The unspecific left ventricular hypertrophy frequently found in patients with hypertension, hypertrophic cardiomyopathy and valvar disease, frequently leads to QTc prolongation. In these patients, even though left ventricular hyperthrophy (LVH) is a risk marker in itself, the QTc has proven to be an additional independent risk marker in multivariate analyses [37]. A previous report demonstrated that surgery may lead to transient QTc prolongation, however, in most of the described cases in this study, changes in electrolyte level or newly added QT prolonging medication could explain the phenomenon [38].

Nevertheless, anesthesia in patients with documented QTc prolongation is challenging, related to the appropriate use of medications.

QTc prolongation as a special risk marker in myocardial infarction was described already 1978 [39]. Both STEMI as well as NSTEMI may lead to QTc prolongation and occurs in some studies in 20–30% of the patients [40,41]. Several studies seem to confirm prolonged QTc as an independent risk marker for sudden cardiac death after myocardial infarction [40–42]. The treatment of myocardial infarction with PCI and medical intervention has improved since 1978. The efficacy of beta-blocker treatment in myocardial infarction, one of the earliest documented treatments, has lately been questioned and is no longer recommended in the treatment guidelines for patients with preserved ejection fraction. The indication of beta-blocker treatment of ACS patients with QTc prolongation seems, however, logical and deserves to be analyzed in currently running studies.

Various scoring systems attempt to estimate mortality risk in patients with marked prolonged QTc time [31,43]. A common feature of these scoring systems is the summation of electrolyte disturbances and number of QT prolonging drugs. We used in our hospital the "QT mortality score", and found that the patients with a QTc mortality score of three or more had a mortality of more than 50% in a three-year period, indeed a very serious prognosis [31]. Studying a cohort with that serious prognosis raises the question of if the QTc prolongation is mainly a marker of very severe disease. However, we judge our data on the influence of QT prolongation on short-term prognosis, as a sign of the additional independent risk caused by QT prolongation [44]. Our interpretation of these data is that a QTc prolongation in the acute face is an additional risk factor and one should intervene.

### 8. Summary

In summary, on the background of current knowledge regarding prolonged QT, and our own well-documented experience, prolonged QTc is a frequent finding on ECGs with computer heart rate corrected QTc. The finding of QTc prolongation should not be regarded as a harmless random finding. Any QTc prolongation above 480 ms represent an increased mortality risk and should be treated appropriately, including correction of electrolyte disturbances and stopping QT prolonging medication. Genetic testing for LQTS-related genes is emerging as a part of a standard for good clinical care.

**Author Contributions:** All authors have read and agreed to the published version of the manuscript.

**Funding:** This research received no external funding.

**Conflicts of Interest:** The authors declare no conflict of interest.

### Abbreviations

| | |
|---|---|
| ECG | Electrocardiogram |
| QT interval | Time in milliseconds between beginning of Q-wave and end of T-wave |
| QTc | QT interval adjusted for heat rate |
| cLQTS | Congenital long QT syndrome |
| aLQTS | Acquired long QT syndrome |
| LAE | Life-threatening arrhythmic event |
| LVH | Left ventricular hyperthrophy |

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
