# Peer review of "Moderately Prolonged QTc in Computer-Assessed ECG, Random Variation or Significant Risk Factor? A Literature Review"

_cardiogenetics, doi:10.3390/cardiogenetics12030025_

Round 1
Reviewer 1 Report
This review entitled “moderately prolonged QTc in computer assed ECG, random variation significant risk factor? A literature review” reviewed the recent publications about the potential mechanisms of QT and clinic determination based on ECG. This is an interesting topic and worth to publication. The following questions and comments need to be fully addressed prior to publication:
1. The introduction is very long compared to other parts, several paragraphs in the intro part can be further discussed in the main text into few sections, for example: a section can discuss the potential genes that associate with QT, another one can summarize the potential diagnose strategy and their own dis- and advantages.
2. The original figure legends from the original referenced papers are not necessary to show, please re-edit the figures and delete the old legends.
3. An abbreviation table is required for explaining specific terms like QT, ECG.
4. In addition, this manuscript needs careful editing in English to make the paper clear to the reader.
Author Response
Dear Editor, Thank you for valuable evaluation of our manuscript and for giving us the opportunity to revise and resubmit the manuscript which hereby certainly has improved. Please, find below our point-by-point response to comments from the reviwers. Skien 1 August 2022 On behalf of the authors, Sincerely, Jan Hysing Reviewer 1 This review entitled “moderately prolonged QTc in computer assed ECG, random variation significant risk factor? A literature review” reviewed the recent publications about the potential mechanisms of QT and clinic determination based on ECG. This is an interesting topic and worth to publication. The following questions and comments need to be fully addressed prior to publication: 1. The introduction is very long compared to other parts, several paragraphs in the intro part can be further discussed in the main text into few sections, for example: a section can discuss the potential genes that associate with QT, another one can summarize the potential diagnose strategy and their own dis- and advantages. Authors’ reply: The Introduction has been split in sub paragraphs with following headlines:
Molecular genetic studies
Mapping of gene penetrance variation
Diagnostics and treatment
LQTS incidence is highly dependent on the cohort studied. 2. The original figure legends from the original referenced papers are not necessary to show, please re-edit the figures and delete the old legends.
Authors’ reply:
The original figure legends have been removed from figure 1 and figure 2. In figure 3 and 4 the original legends is such an integrated part of the figure that deleting this will reduce information in the figure. 3. An abbreviation table is required for explaining specific terms like QT, ECG. Authors’ reply: A list of abbreviations has been added 4. In addition, this manuscript needs careful editing in English to make the paper clear to the reader. Authors’ reply: The manuscript has been reviewed and reviced by an English language expert

Reviewer 2 Report
Authors reviewed data concerning QT measurement.
The review is well-written. Some minor points should be clarified:
1.- Please include data focused on genetics (main genes) but no data included concerning minor genes associated with QT. Any differences in enlargement of QT interval?
2.- Please include data concerning genetic variants (common variants) and susceptibility to enlarge QT interval.
3.- Please include novel data concerning aim of the review. The most recent publication included in the manuscript is 2019.
4.- Please highlight the idea that despite computer help in QT measures, all parameters should be revised by an expert cardiologist before adoption of treatment.
5.- Please include any data concerning ages. Any differences between children adults and old population?
Author Response
Dear Editor, Thank you for valuable evaluation of our manuscript and for giving us the opportunity to revise and resubmit the manuscript, which hereby certainly has improved. Please, find below our point-by-point response to comments from the reviwers. Skien 1 August 2022 On behalf of the authors, Sincerely, Jan Hysing Reviewer 2 The review is well-written. Some minor points should be clarified: 1.- Please include data focused on genetics (main genes) but no data included concerning minor genes associated with QT. Any differences in enlargement of QT interval? Authors’ reply to 1 and 2: A new paragraph has been added under the headline: Molecular genetic studies: The errors in the three ion channels could be linked to the clinical syndromes LQT1, LQT2 and LQT3. The genes leading to errors in these channels contribute to 75% of all clinical cases, and are now characterized as major LQTS genes [10]. Efforts to explain the remaining heritability of LQTS led to reporting of numerous new genes linked to clinical syndromes labeled LQT4 to LQT17. These genes have now in recent years been characterized as minor LQTS genes. The gene-disease associations for many of these minor genes seems to be questionable, and their effects seems easier to explain in a poly genetic model. In polygenic models even common gene variants may contribute to development of rare clinical syndromes. These polygenetic models also give support to the term “reduced repolarization reserve” originally introduced by Roden [11] in 1998. 2.- Please include data concerning genetic variants (common variants) and susceptibility to enlarge QT interval. Authors’ reply:see above 3.- Please include novel data concerning aim of the review. The most recent publication included in the manuscript is 2019. Authors’ reply: Novel data has been included, the paper now cites references from 2020, 2021 and 2022 4.- Please highlight the idea that despite computer help in QT measures, all parameters should be revised by an expert cardiologist before adoption of treatment. Authors’ reply: The text has been revised as follows Certainly, the computer corrected ECG must be manually checked for electromechanical noise and arrhythmias such as extrasystoles and bigeminy. Before making the diagnosis of congenital LQTS the ECG should be checked by cardiologist 5.- Please include any data concerning ages. Any differences between children adults and old population? Authors’ reply: The text has been revised with an additional paragraph as follows
Congenital LQTS tend to be exposed clinically in childhood or adolescence [22]. Pediatric ECGs is a special challenge because of the physiological tachycardia in small children. ECG with heart rate more than 100 should be judged with caution related to the substantial Bazett’s correction. Aquired QTc prolongation more often are found in patients after 40 years of age and predominant after the age of 60 years.
